# Exploring Individual Mental Health Issues: A Qualitative Study among Fellowship-Trained Sports Medicine Physicians

**DOI:** 10.3390/ijerph20075303

**Published:** 2023-03-29

**Authors:** James Stavitz, Adam Eckart, Pragya Ghimire

**Affiliations:** 1Graduate College of Health Professions and Human Services, Kean University, Union, NJ 07083, USA; 2Athletic Training Program, Kean University, Union, NJ 07083, USA; 3Exercise Science Program, Kean University, Union, NJ 07083, USA

**Keywords:** qualitative research, public health, emotional well-being, behavior, athletics

## Abstract

The mental health of fellowship-trained sports medicine physicians (FTSMPs) around the United States is a subject that needs additional exploration. Currently, there is little research exploring how FTSMPs address their mental health on a routine basis. Using the theory of secondary trauma stress to help navigate this study, the purpose of this expressive, all-purpose qualitative study is to improve the understanding of FTSMPs’ perceptions of their mental health and the kinds of strategies used to manage these issues. This is a general qualitative study. All interviews were conducted via video communication platforms such as Zoom. The final sample included 35 FTSMPs: 25 men and 10 women. Data collection used a semi-structured interview approach. Data analysis was carried out using NVivo 12 qualitative data analysis software. Four themes emerged: mental health matters affect individual daily lives of FTSMPs; FTSMPs correlate mental health struggles with stress and anxiety; FTSMPs experience barricades when seeking support for mental health issues; and FTSMPs have poor mental health support-seeking behaviors. Results highlight openings for hospitals and private practice institutions, including producing a maintainable work–life equilibrium for FTSMPs and offering these FTSMPs access to mental health services. These recommendations may diminish exhaustion amongst several FTSMPs, a product detrimental to patients, providers, and hospitals.

## 1. Introduction

The mental health perils of fellowship-trained sports medicine physicians (FTSMPs) have become prominent with the dawn of the coronavirus disease 2019 (COVID-19) pandemic. These dangers include emotional anguish, sleeplessness, alcohol/drug abuse, indications of posttraumatic stress disorder (PTSD), hopelessness, apprehension, fatigue, antagonism, and higher perceived stress [1]. The pandemic’s toll on the mental health of FTSMPs and frontline healthcare workforces remains unclear, specifically concerning fatigue because of heavy work capacity [2,3,4]. However, fatigue is not the only mental health question facing healthcare supporters [5].

When a physician is “fellowship-trained”, this means that they have participated in a fellowship program to become a specialist in their chosen field. During that program, a physician will shadow a qualified specialist, training with that specialist to provide care in that particular field. For example, a physician may already be an orthopedic specialist. Still, if they would like to become an expert in sports medicine, they would complete a fellowship training program in sports medicine. A fellowship program can require a doctor to participate in hundreds of focused surgeries before they obtain their credentials [6].

Granting that COVID-19 has affected the indirect suffering experienced by all healthcare providers, it is far from the sole cause [2]. Relatedly, FTSMPs often advance empathy exhaustion or a failure to communicate to patients with sympathy because of the constant understanding of poor patient outcomes or vulnerability that has occupied them [5]. These and other matters that may deteriorate their mental health are faced by FTSMPs, to some extent, on an everyday basis [7]. Accordingly, in sports medicine, it is necessary to appreciate these physicians’ perception of their mental health and the varieties of tactics they use to improve their mental well-being. The theory of secondary trauma stress (STS) was established by academics, including Stamm (1995), to clarify the toll that care delivery can take on the mental well-being of those who work with people who have gone through alleged suffering, such as an injury while participating in a sport [8]. The STS model proposes consistent exposure to others who have faced hurtful strain perpetrates a consequential form of traumatic stress [8]. Ludick and Figley (2017) note a sequence of 12 influences that govern the degree to which persons working with patients who have suffered an injury or are in search of medical help experience secondary traumatic stress or are unaffected (Table 1) [9].

This 12-factor model incorporates factors linked with sympathy, strength, and secondary stress. Accordingly, the STS model proposes a framework within which those tactics can be understood and supports this exploration. Unrestricted investigation of the strategies permits them to be positioned within the paradigm, heightening our comprehension of the connection between STS theory and practice or categorizing central issues that arise in practical stress management for which the template presently does not deliver support [9].

The mental health of FTSMPs around the US is a subject that needs additional exploration. Currently, there is little research exploring how FTSMPs address their mental health on a routine basis and their experience and perceptions regarding their mental health. Using the theory of secondary trauma stress to help navigate this study, this expressive, qualitative study aims to improve the understanding of FTSMPs’ perceptions of their mental health and the strategies used to manage these issues.


**Research question: What are FTSMPs’ views of understanding and treating their mental health issues based on lived experience?**


## 2. Material and Methods

To address this question, this exploration, which started in spring 2022 and ended in fall 2022, was directed using an informative, conventional qualitative research model [10]. This study attempts to provide an exploratory picture of the research matter rather than search for greater profundity regarding the understanding of the individual contributors.

Fellowship-trained sports medicine physicians were the targeted population who treat and interact with athletes regularly. The inclusion measures included the following criteria: (a) currently practicing as an FTSMP, (b) having a broad understanding regarding their own mental health, (c) currently employed as a sports-team physician at any level, (d) actively treating athletes on a routine basis, and (e) must be fully fluent in the English language. These inclusion measures were designated out of an amalgamation of safeguarding the contributors’ identities and ensuring the study was ethical and feasible.

Criterion and snowball sampling identified United States (US) FTSMPs by contacting hospital directories across the country and filtering the providers that are FTSMPs [11,12]. Fellowship-trained sports medicine physicians’ contact data can be found on the physician practice website and is made available to the public. To confirm that all likely participants met the inclusion standards, the research team established a prescreening survey via Qualtrics™, asking participants to answer precisely regarding their inclusion standing. This invitation is included in the letter of solicitation (LOS) sent to each possible contributor, along with the link to the Qualtrics™ survey. If contributors answered “yes” to all inclusion requests and were interested in contributing to the research, they were directed to provide an email address to plan a meeting and submit the Qualtrics™ survey. Five hundred FTSMPs were contacted by sending the LOS with a link to a prescreen examination that stipulated the inclusion criteria. Of the 500 potential participants contacted, 75 prescreening surveys were started, and 35 were completed.

The closing sample involved 35 participants, comprising 25 men and 10 women. The final sample was presently practicing across the US. It embodied various FTSMPs and their respective medical experiences while endeavoring to achieve steady work, family life, and patient caseload (Table 2). The presence of FTSMPs widens the conceptualization of sports medicine practice outside that which many academics discuss, giving the study significant originality.

Data collection occurred over six months. An interview was scheduled after the signed informed consent document was accepted, which was sent via email to the participant before the interview. Interviews were conducted virtually over the Zoom online platform. Before the discussion, a verbal agreement to begin the study, collect data, and record the interview was approved. An interview manual was organized to help answer the primary research inquiry and exploratory questions to discuss any new notions raised during the interview (Table 3). The research team asked two collaborators with 15 years of qualitative research practice each to review the interview guide for comprehensibility, dependability, and consistency [13]. The PI, who has five years of qualitative research practice, completed all the interviews. Interviews lasted 30–60 min, were audio was logged, then later transcribed precisely. Participants were asked to member-check the transcriptions. Figure 1 exemplifies the interview procedure, and Figure 2 illuminates the steps used for data gathering, starting with the Institutional Review Board (IRB) consent.

Data analysis was completed using the NVivo 12 qualitative data analysis software to form and accumulate codes. After each interview was transcribed and verified, it was uploaded into NVivo 12 to initiate the coding process. Codes were established from the evolving data. This means that the research team did not go into the analysis phase with a predetermined list of codes but instead let the codes come from the data itself. The constant comparative method was utilized for coding [14]. Coding occurred concomitantly with the interviews, so the research team was still questioning participants as the initial coding began until saturation was attained. No further themes appeared from the data assembly [14].

The codes were continuously linked to the new data through constant comparative coding to ensure the code implications were unswerving, and modifications were made to the codes as needed [14]. If modifications were made, the research team would return to the preceding discussions and determine whether the codes aligned. Early on, the research team utilized line-by-line coding, and initial codes involved a few words, one or several sentences, or a passage, and 700 codes emerged [15].

The research team followed the six-step thematic data analysis process to achieve confirmability, transferability, and dependability [14,15]. After the PI transcribed each transcript precisely, the research team instituted familiarity by rereading and interpreting each transcript to support the analysis. Next, the research team executed open-and-descriptive coding [14,15]. Once initial codes were created and common groupings were produced, themes were created, and two professional content qualitative experts, who are FTSMPs and have been practicing for over 15 years, read the interview transcripts and analyzed codes for code agreement (agreement verified), to establish validity and reliability, and to account for researcher prejudice.

Next, preliminary themes were confirmed by the research team, judiciously checking each theme alongside the data and confirming that it truthfully embodied the contributors’ ideas. Then, the themes were cross-validated. Finally, the research team accumulated and conveyed the themes (Figure 3). Again, consensus-reaching was established to interpret the findings and address the research questions [11,12,13].

Vigilant efforts were made to preserve transparency and trustworthiness. Foremost, the *DoCTRINE Guidelines: Defined Criteria to Report INnovations in Education* was used to improve the value and transparency of this research (Table 4) [16]. This study was aligned in all mechanisms to confirm that the results addressed the research question. This involved thoroughly endorsing the interview guide with qualitative specialists before the interviews were performed. Next, member-checking was utilized. Participants were allowed to evaluate and amend their interview transcripts, ensuring that the interviews perfectly captured the participants’ experience and perception [17,18].

Bracketing was utilized to reduce the effects of researcher bias [19]. The next step of the analysis also served to remove prejudice by ensuring that every theme could be truthfully sourced in the data. Evidence of this sourcing of ideas can be located using candid quotes from contributors in the results and discussion sections [20].

Credibility was achieved by conducting peer debriefing to offer a peripheral check on the research development and member-checking to assess the discoveries and interpretations with the participants. The research team provided plentiful descriptions so that those who seek to transfer the findings to their site can critique transferability [21]. The research team attained dependability via audit trails. Scholars have noted that a study and its outcomes are “auditable” when another scholar can undoubtedly follow the decision trail [22]. Koch (1994) argued that another investigator with comparable data could reach comparable, but not opposing, conclusions [23]. The research team reached this by keeping records of the raw data, field notes, transcripts, and an insightful diary to help systematize and cross-reference information. Confirmability was established when credibility, transferability, and dependability were achieved [23]. Researchers have stated that trustworthiness is constructed throughout the interview [24].

Audit trails consist of physical and logical audits. Logical audits include record keeping of the researcher’s reflections and thought developments during the process. Physical audit trails consist of all the manuscripts used in the research and trace the choices and approaches used [25]. Harrison, MacGibbon, and Morton (2001) note that audit trails are valuable and effectual in augmenting the trustworthiness of research such as this [26].

Contributors who conveyed that they are members of a population designated by the IRB as at-risk or vulnerable have been omitted from this work, so no members of vulnerable populations were involved in this study. IRB support was attained before participant enrollment began, and all contributors signed the informed agreement form before any data was collected.

Once the interview was set, the participant was given a code name. This code name is the lone way to link the interview with the prescreening data to ensure privacy [20]. The research team recruited participants with whom they had no other relationship.

Additionally, participants’ identities have been kept confidential. Interview transcripts have been de-identified by deleting participants’ names and other identifiers, including but not limited to the name of the employing facility and its precise geographic location. Pseudonyms are substituted for the participants’ names in all work products linked to the research [18].

## 3. Results

Four themes emerged from the thematic analysis of the data (Table 5). Table 6 shows examples of robust, direct quotes from the participants that helped create each theme.

Theme 1: Mental health matters affect individual daily lives of FTSMPs.

Of the participants, 30 FTSMPs detailed that “mental health” was tough to label; however, based on personal understandings, most contributors professed that mental health is a facet of one’s natural life that affects one’s fitness to execute everyday responsibilities. P15 extended that a mentally healthy individual is “able to reason [undoubtedly].” P20, in the same way, stated that mental well-being involves one’s “present state of mind” to be able to think evidently “and without predisposition.” P22 mentioned prejudice, as she felt that a person with mental health problems leans toward being “emotional when making decisions.” P11 had parallel views and stated, “If you reflect about your most rationally clear moment, the whole moment was healthier. Everything made sense. If you think about a moment [when], you were emotionally unwell, nothing seems to drop into place.” P23 alleged that mental lucidity impacted his decision-making, which, in turn, influenced his patients’ results. P9 clarified, “When I am emotionally and spiritually apt, I tend to apply more effort concentrating on my work. I notice a big difference when conscious about my mental health.” P29 and P18 thought that, as FTSMPs, they could not commence providing services to others if they did not support their own physical, mental, and emotional well-being.

In addition, P16 detailed that a person with [mental] health problems may also have advanced physical [health] issues and lack a community life. P24 explained, “[Mental health] improved the societal facet of my life. It is easier for me to relax, connect, and intermingle better.” P2, P5, P10, and P21 alleged that being psychologically fit meant they had optimistic behavior and proper interindividual capabilities. P16 explained that in moments when she had “fewer mental health issues,” she was not “always experiencing harmful thoughts”.

With mental health issues linked with physical, emotional, social, and cognitive well-being, the contributors commonly alleged that facing these issues could distress one’s productivity in ordinary life. P19 expanded, “Mental health contains our expressive, spiritual, and social well-being. It marks how we reason and express [ourselves]. It supports how we control stress from trauma, connect to others, and adapt our life choices.” P1 held that since mental health issues are associated with experiences, dealing with mental health “is a multifaceted” treatment model.

The results propose that the FTSMPs’ view of dealing with mental health encompasses a customized methodology with the impartialness of refining one’s total well-being and working life. Unambiguously, the contributors may have suffered mental health troubles due to the absence of management skills relative to their specific issues. Thus, the lack of coping skills may distress one’s decision-making and endanger their ability to treat their patients properly.

Theme 2: FTSMPs associate mental health struggles with anxiety and stress.

The FTSMPs defined different areas of their lives that cause stress and, consequently, influence their mental health. As pronounced by the participants, the sites of life linked with mental health issues were patient care, busy schedules, family, and finances. Most participants were most strained about stabilizing their family’s schedule, work, individual schedule, and social life. Of the contributors, 22 felt pressures related to the high volume of their patient caseload. As FTSMPs, many participants were stressed by the conjecture that their patients relied on them to be mentally strong. P13 shared that patients saw sports medicine as “an occupation” rather than “a person with [their] own issues.” P3 expressed, “People come to me for assistance…Yet, many of the competitors we see look at us as “services” and don’t understand that we are individuals with our issues”.

The contributors perceived that they needed work–life stability. P20 faced worry from managing a mixture of “problematic” patients, busy individual schedules, and trying to be an active part of his family. The participant defined “problematic” as issues he may not seem to answer. P10 reported that he overstretched himself to the extent of having his employment “capture” his life. P2, P5, P6, and P9 also experienced stress from their job “overwhelming” their lives. P14 shared, “[I often missed out on] life events and commitments due to the high request of the job.” P27 noted, “I [infrequently] find the time to unwind.” P5 also felt that she was missing out on life, particularly during the COVID-19 pandemic, and noted that she thought all she could do was “either work or be at [her] clinic and had no time” for herself. P5 stated, “My main period with mental health issues is when I am placing much labor into my work, and I am burnt, but there is no fulfillment”.

A number of 27 participants noted that they experienced mental health matters from the pressure of their families because they feel that they give all their time to their employment. P3 highlighted that his apprehension came from tasks that increased as he advanced throughout his career and became more involved due to his work experience. P13 described her typical night after her long day when her child and dog kept her up. P13 stated, “I have a long day of clinical rotations, and just knowing this adds to the stress.” Additionally, P13 was the only participant who cited the stress of being overtasked and felt underappreciated. The participant shared, “I am attempting to… [build] for my future, but I am not seeing a lot right now. 60 to 70-h weeks, not being reimbursed nearly enough, and dealing with trying patients is starting to get to me”.

A minority of participants mentioned that their stress and mental health issues began while studying to become healthcare professionals. P11 experienced “fatigue” and anxiety while studying. P2 explained that the pressure in school came from stressing himself to “thrive in a very aggressive academic institution,” which he feels may have carried over into his career.

Theme 3: FTSMPs experience barricades to looking for support for mental health issues.

The participants faced extrinsic hindrances to looking for help for mental health. External barriers were their responsibilities to their communities, public view, insurance, other expenditures, and finding a mental health professional. Most participants communicated that, as FTSMPs, they were torn between their work and personal lives. P17 stated, “The biggest barrier, I would say, is trying to balance and assess a busy calendar, work, societal expectations, family as well as staying psychologically and emotionally clear.” A number of 28 contributors thought the same and noted that they have difficulty finding a moment in their schedule to make an appointment with a mental health professional.

A number of 15 subjects in the study described that their health insurance did not cover mental health and that they had to pay for the services out of pocket. Further, 11 subjects explained that finding an individual with whom they were content was also a barrier. P23 noticed that being employed as an FTSMP meant that the health professionals in their local area knew her, which made it more difficult for her to seek help.

A number of 17 participants mentioned that mental health issues seem to be branded and stigmatized in their local community. P31 and P34 described mental health issues as “vilified.” P16 explained that mental health was quickly discussed in seminars as a student. P8 stated, “We talked about mental health occasionally, but it seemed we had more important things to discuss at that time”.

Theme 4: FTSMPs have poor mental health support-seeking behaviors.

Contributors to the study made it known that they had never pursued mental health services or had sporadically gone to see their mental healthcare provider. Of participants, 13 said they had visited a mental health professional when they were children. However, P4 recalled, “As a child, my mother and father made me [see a mental health therapist]… In my grown-up life, [I’ve] spoken to a few, but very randomly and inconsistent with their recommendations.” Among the 35 participants, only P34 continued to see her therapist “simply for maintenance habitually.” P34 reported that her whole family had been “good at taking care” of their mental health.

A few participants noted that they have yet to pursue help for their mental health. P7 and P9 felt they were “okay” and did not need help. P3 thought he “certainly could use some guidance” to help him in certain stressful situations. P2, likewise, believed that he could utilize the support of mental health professionals even though he never sought one out; P2 added that he performs routine self-care acts such as “getting a decent night’s sleep, eating healthier… [to] progress not only my mental health but physical also”.

P1, P5, P6, and P8 pursued specialized mental health care when faced with issues as an adult but did not habitually follow up on their visits or the medical recommendations from their providers. The contributors had comparable experiences of apprehension when they were in high school. P5 sought treatment after she experienced a “disturbing occurrence” when she was younger but sought “no upkeep.” As an adult, she “went to treatment on and off” and felt that she had learned coping aids needed for taking care of her mental health. P8 stated that she is often “too busy” as a scholar to find the time to pursue a mental health professional. She did not comply with the recommendation to see her therapist on the suggested weekly schedule, which she expressed added to her angst. P6 works in a teaching hospital and noted, “Academic stress would give me panic and anxiety attacks,” and she became depressed. The participant described an instance during which she had to pull over while driving, as she thought she was “going to have a stroke” when her anxiety and depression became very evident. The participant noted that recently, she visited her therapist “once every few months” but acknowledged that she “needed more visits”.

## 4. Discussion

### 4.1. Theme 1

Based on the corresponding literature regarding FTSMPs, mental health proved to be a substantive and prevalent issue for the contributors in this study. The first key theme echoed mental health’s significance and importance. Mental health was not only suspected of contributing to the participants’ ability to practice, but every aspect of their life was impacted in various ways. Rather than concentrating on mental health struggles, the subjects focused on a more all-inclusive sense of mental health with prominence placed on its expressive, societal, and intellectual components. The concepts discussed, in this regard, are more similar to those defined in the literature as “well-being” than necessarily “mental health” [27].

Collectively, participants identified the role of preserving their mental health in their ability to provide the necessary attention to their patients. This idea has support within the current literature, such as from efforts to combat burnout. Furthermore, the participants’ attention to their social affairs concerning mental health is associated with Gibbons et al.’s (2019) findings linking salary, job satisfaction, quality of life, and mental health in healthcare providers [28].

### 4.2. Theme 2

Even though the subjects did not concentrate on complex issues in their overall dialog, mental health issues were a practical aspect of the second key theme. Most contributors described experiencing mental health issues from stress, mainly related to finances, work, individual schedules, and family. Work, patient struggles, high volumes of patients, and finances were common causes of stress and mental health issues for FTSMPs. One participant, P3, indicated that the student athletes they could not help were their primary source of work-related anxiety. However, no other contributors meaningfully addressed secondary trauma as P3 did. In its place, their work-related stress and resulting mental health struggles stemmed most often from work–life stability.

### 4.3. Theme 3

Work–life balance is a common issue in many healthcare fields, including sports medicine [29]. Even those contributors who did not confirm such stress unswervingly pointed to family issues as a catalyst for stress at work. Several subjects did note the deterioration of their mental health due to the COVID-19 pandemic. The past literature has highlighted the toll of the pandemic on healthcare professionals (1). Yet, in this study, the results linking to the pandemic did not automatically narrate the amplified stresses of sports medicine and healthcare provision so much as the pandemic’s negative effect on their preferred ways of handling stress, such as travel.

The current research outcomes relate to the drivers of FTSMPs regarding their experiences and perceptions of personal mental health issues. Not only was an absence of work–life balance a dominant challenge, but also the consequences of the COVID-19 pandemic on the contributors varied significantly. These outcomes, consequently, offer a vital novel contribution to academics on the mental health of FTSMPs.

### 4.4. Theme 4

The outcomes of this research offer awareness of why those problems are complicated to deal with. The struggle between work and livelihood persisted as a prevalent barrier when looking at experiences and perceptions of personal mental health needs. FTSMPs striving to find time for work and personal necessities meant that looking for care from a healthcare professional would signify an untenable burden in their already busy schedules. In this respect, there was no instance found in past research for the focus on the work–life balance among FTSMPs while concurrently seeking medical care as an obstacle to health care in professional medicine.

Additionally, it is concerning that sports medicine professionals lack sufficient insurance for mental health care. Despite spending numerous years in academia and preparing to become physicians, many participants noted the stigma of humiliation when seeking mental health care. These challenges are usually addressed through education and instructive courses, so their noticeable presence in this study is troubling (28). Given these hindrances, it is unsurprising that the contributors in this research noted an overall poor level of application of mental health care services. Participants were divided regarding whether they realized they needed professional care. However, the occurrence of this barrier and lack of apparent accessibility of care was noteworthy enough that even those who did not feel they needed these services could have figured this to be, in part, the reason that they alleged such care was something uncommon for individuals in their position as FTSMPs [29].

### 4.5. Theory of Secondary Trauma Stress

As the sports medicine industry continues to implement practices to help improve mental health and well-being amongst FTSMPs, we need to ask ourselves [4,5,26,27,28,29]: “Why is there a need to improve these practices?” Is the purpose to bring awareness, help with decision-making, or promote change? These factors can help decrease the divide between what is understood and what is undertaken. With mental health, health experts have seen over the last few years how our data may cause a decisive shift in the actions taken [4].

Therefore, for this exploratory research, the theory of secondary trauma stress (STS) is appropriate to help guide this research. As discussed earlier, STS notes 12 variables that link sympathy, strength, and stress. Of these 12 variables, *other life demands, compassion fatigue resiliency, self-care, detachment, sense of satisfaction, and social support* were used to help guide this research and provide a lens to help better understand FTSMPs’ experiences and perceptions when dealing with personal mental health issues.

FTSMPs nationwide can utilize the knowledge found in this study to create a safer environment for themselves, their families, and their patients. The collected information provides insight into barriers regarding experience and perceptions of personal mental health issues. Now that we better understand this topic, we can move forward in other research to address the daily barriers and challenges faced by FTSMPs and other sports medicine professionals.

One weakness of the current research is its dependence on the honesty and truthfulness of participants’ reports of their perceptions and experiences. While there will be no way to confirm that participants have given authentic and truthful data, participants were guaranteed that their identities would remain private to foster reliability. Furthermore, participants were asked to analyze and review the transcripts of their interviews and recommend any modifications that would make the data more precise. A second limitation is that it cannot be known whether factors irrelevant to the research are influencing participants’ answers during the data collection process. Asking participants to member-check, analyze, and correct their transcripts is a deliberate choice to decrease the likelihood of biases, which may be due, for example, to temporary exclusive distresses.

Another limitation is the sampling of the study. This sampling was not random and may not entirely represent the whole population, and the results will only be generalizable to some FTSMPs. In addition, more men than women participated in this study, and most contributors were from New Jersey, New York, Pennsylvania, Delaware, Florida, Texas, and California.

Upcoming exploration may address the following:Expand on this current study to look at a larger population outside FTSMPs.Utilize the current study using quantitative measures and hypothesis testing to predict a possible outcome.Involve an interventional study to improve the findings regarding challenges and barriers.Explore educational advancements that may impact the challenges and barriers regarding mental health related to sports medicine, such that healthcare professionals can improve experiences and perceptions for themselves and their patients.

## 5. Conclusions

This study sought to answer one fundamental research question: What are FTSMPs’ lived experiences and perceptions of dealing with their mental health issues? Analysis of interviews with 35 FTSMP in the United States showed that poor work–life balance is a fundamental cause of mental health struggles amongst these homogeneous populations. Numerous contributors also cited secondary trauma and empathy exhaustion.

Although COVID-19 may have exposed issues regarding work–life balance, it was not the solitary cause of them. Instead, the pandemic deteriorated participants’ mental health by preventing them from stress-relieving leisure activities that the study contributors participated in before the pandemic. At the same time, participants also noted a highly problematic level of stigma regarding mental health. Many did not have adequate insurance coverage that was in-network with what they perceived as appropriate mental health care. Overall, utilization of mental health care services was low.

The study’s outcomes have strong inferences regarding how FTSMP employers can and must do more to support themselves, their families, and their patients. Failure to produce a maintainable work–life balance and offer appropriate access to mental health care may lead to enhanced FTSMP burnout. This consequence is very harmful not only to the provider themselves, but also to patients and medical institutions alike. However, the results of this study advocate that private practices and hospital systems can fundamentally and applicably ameliorate this issue through increased staffing and insurance plans that cover mental health services.

## Figures and Tables

**Figure 1 ijerph-20-05303-f001:**
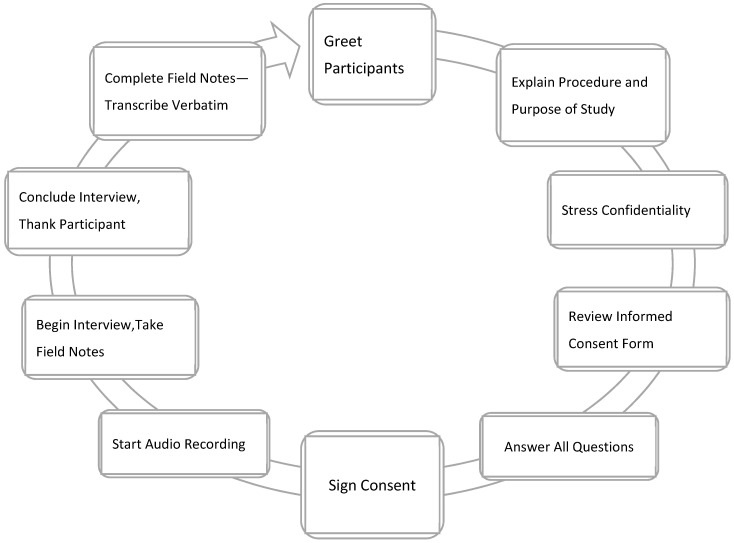
Interview Process.

**Figure 2 ijerph-20-05303-f002:**
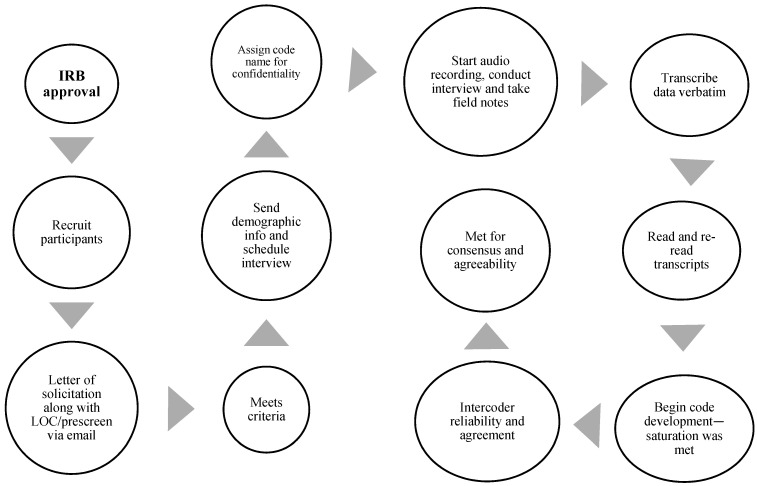
Data collection.

**Figure 3 ijerph-20-05303-f003:**
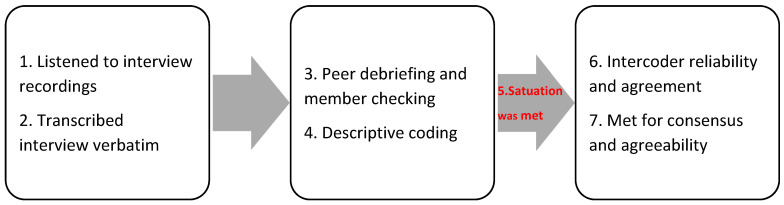
Data analysis.

**Table 1 ijerph-20-05303-t001:** STS 12-Factor Model.

(1) Exposure to suffering;
(2) Empathic concern;
(3) Empathic ability;
(4) Empathic response;
(5) Traumatic memories;
(6) Prolonged exposure to suffering;
(7) Other life demands;
(8) Compassion fatigue resilience;
(9) Self-care;
(10) Detachment;
(11) Sense of satisfaction; and
(12) Social support.

**Table 2 ijerph-20-05303-t002:** Participant Demographics.

Gender	Percentage	Number of Contributors
Men	71.42%	25
Women	28.51%	10
Prefer not to say	0%	0
**Location:**		
California	5.71%	2
Colorado	2.86%	1
Connecticut	11.42%	4
Delaware	11.42%	4
Florida	17.14%	6
New Jersey	20.00%	7
New York	14.28%	5
Pennsylvania	5.71%	2
Rhode Island	2.86%	1
Texas	5.71%	2
Virginia	2.86%	1

**Table 3 ijerph-20-05303-t003:** Interview Questions.

Tell me a little bit about yourself:
2. Why did you decide to become an FTSMP?
3. What do you know about mental health?
4. As an FTSMP, describe a time, if at all, that you experienced issues with mental health:
5. As an FTSMP, describe a time, if at all, that you had to seek professional support to help you deal with a personal mental health issue:
6. As an FTSMP, what is your biggest challenge when dealing with personal mental health?
7. As an FTSMP, what barriers may you experience when dealing with personal mental health?
8. As an FTSMP, what are some pressures that you may face when dealing with personal mental health?
9. As an FTSMP, what are some benefits to dealing with your mental health issue?
10. If you can remember, has your mental health improved or regressed since becoming an FTSMP?
11. As an FTSMP, how can you improve your mental health?
12. Are there any other questions or comments you would like to make regarding mental health, mental health as a healthcare professional, or anything?

**Table 4 ijerph-20-05303-t004:** Defined Criteria to Report INnovations in Education (DoCTRINE).

Introduction	Y/N
Need for the curriculum	
2. Review of the relevant literature, theories, models, or published curricula	
3. Unique contribution of the curriculum to the literature	
Curriculum Development	Y/N
4. Purpose/goals of the curriculum	
5. Outcome-based learning objectives	
6. Target population of learners	
Curriculum implementation	Y/N
7. Instructional setting for curriculum delivery	
8. Resources for implementing the curriculum	
9. Description of instructional methods	
10. Methods to evaluate achievement of outcome-based learning objectives	
11. Origin of evaluation instrument(s)	
Results	Y/N
12. Number of learners participating in the curriculum	
13. Number of participants included in the evaluation	
14. Evidence of achievement of outcome-based learning objectives	
Discussion	Y/N
15. Summary of findings	
16. Interpretation of findings in relation to the existing literature	
17. Lessons learned from the implementation of the curriculum	
18. Limitations of the evaluation of the curriculum	
19. Describes future implications of the curriculum	

**Table 5 ijerph-20-05303-t005:** Themes.

	Number of Transcript Excerpts Included	Number of Participants Contributing
(1) Mental health matters affect individual daily lives of FTSMPs.	87	28
(2) FTSMPs correlate mental health struggles with stress and anxiety.	43	25
(3) FTSMPs experience barricades when looking for support for mental health issues.	65	33
(4) FTSMPs have poor mental health support-seeking behaviors.	72	33

**Table 6 ijerph-20-05303-t006:** Example of Robust Quotes Used to Make Themes.

Theme	Quote
Theme 1	**Q: As an FTSMP, describe a time, if at all, that you experienced issues with mental health:** P9: “Yes. Absolutely. All the time. I have anxiety multiple times throughout the day. I feel like I’ve had it for a while, but I also feel like it has worsened as I’ve gotten older. Perhaps it’s because of more responsibilities. More people relying on you.” **Q: As an FTSMP, what is your biggest challenge when dealing with personal mental health?** P10: “The biggest challenge I would face probably is time and the perception of seeking help. Where am I going to find the time daily to have these conversations? I work, I have kids, have a family. Where’s the time? My only real time is early morning, and no one is available. I can take off work once a week or twice a week or every other week because of too many responsibilities at work. As far as public perception is concerned, but face the facts, unfortunately, mental health is seen as a weakness in some people’s eyes.” **Q: As an FTSMP, what are some pressures that you may face when dealing with personal mental health?** P6: “The most pressure I feel is, I guess, from myself. But I also feel pressure from my job to perform at an optimal level mentally, physically, and emotionally. Not only that but intellectually. People come to me for help. They come to me because they are under the impression that I am mentally fit to do my job, and I am. However, many of the patients that we see look at us as providers and don’t realize that we do have our issues. Every day all of us go home and deal with personal issues. It can be tough. But you know what, you do what you gotta do.”
Theme 2	**Q: As an FTSMP, describe a time, if at all, that you had to seek professional support to help you deal with a personal mental health issue:** P6: “No, I never actually spoke to someone. I probably should have [due to stress]. But no, I never actually saw any professional help. I knew how to get back on track [when dealing with stress and anxiety]. Getting a good night’s sleep and eating better all trickles down to how I can improve not only my mental health but physical and stress. But I can’t comment on what seeking professional help would do because I never actually saw it myself.” **Q: As an FTSMP, what is your biggest challenge when dealing with personal mental health?** P4: “The biggest challenge, I would say, is balancing work, social life, and family, as well as staying mentally clear and on and on top of that, staying mentally on track. It’s easy to derail yourself. I also enjoy staying physically fit, so I believe that can contribute to a boost in my mental health. Sometimes it just makes me feel better to go for a run. I also notice a big difference when I sleep better. I guess it’s cyclical. Yeah, the biggest challenge is balancing everything and staying mentally focused.” **Q: As an FTSMP, what barriers may you experience when dealing with personal mental health?** P2: “I don’t know if I would say I have any barriers. It’s hard to say. Some barriers to my mental health? I would say I am my barrier, probably. I have much anxiety. I strive to be a perfectionist. It is stressful. Sometimes my perfection or perception of perfection can get in the way. Maybe a barrier would be trying to convince me that I need to take a step back before I burn myself out, which I’ve done repeatedly.”
Theme 3	**Q: As an FTSMP, what barriers may you experience when dealing with personal mental health?** P8: “Some barriers, I would say, would be being vocal about a situation. Again, this is my barrier. Some barriers outside of myself? Work, life, trying to find the time. These are all barriers. How am I supposed to find the time when all professional therapists work the same hours as I do?” **Q: As an FTSMP, what are some pressures that you may face when dealing with personal mental health?** P1: “I would say that all the pressures I experience are from myself. Internal pressure. Very little room for error in my profession. If I make one little mistake, it could cost me my whole career. The pressure can be so intense sometimes that I know I should find someone to talk to, but I would rather focus on my craft so these mistakes do not happen.” **Q: As an FTSMP, what are some benefits to dealing with your mental health issue?** P1: “Dealing with personal mental health issues as a healthcare provider is extremely important. Not only for myself. For my patients. I can focus more on my patients and their outcomes when I am mentally fit. It’s easier for me to keep up with my mental health and try to catch up with it. It’s a struggle, though, every day. Every day is a new fight. Finding the time, balance, and the right person for the right situation. Let’s face it, every mental health issue needs extra attention. How are you supposed to find the proper attention? It isn’t easy. But we are all works in progress.”
Theme 4	**Q: As an FTSMP, what is your biggest challenge when dealing with personal mental health?** P1: “I would say my biggest challenge, for me, would be just time. It is hard to find the time. I know that there are not many specialists where I live. Insurance can be an issue as well. Very costly. But yeah, time, finding a specialist, insurance… All are pretty big challenges.” **Q: As an FTSMP, what barriers may you experience when dealing with personal mental health?** P9: “Barriers for me, personally, would be acknowledging that I have an issue when I do. That always has been my struggle—the acknowledgment…The acknowledgment of the issue itself tends to be a significant barrier for me. That’s what I feel is the hardest part. Once I acknowledge the behavior, I must figure out how to deal with it. I do not always make the best decisions with mental health treatment, but I do the best I can.” **Q: As an FTSMP, what are some pressures that you may face when dealing with personal mental health?** P11: “The most significant pressure when dealing with a personal health issue is probably from me. If that make any sense? Do you know? It’s tough. Something that you can’t see, you can certainly feel it. No one else can see it. It can certainly be a challenge. It isn’t easy. How are you supposed to differentiate between being in a bad mood versus being overwhelmed versus being burnt out versus just everyday anxiety? How are you supposed to know? If you don’t know, how are you supposed to get help? Is anxiety normal? I don’t know… It’s not talked about a lot. ”

## Data Availability

To protect participant confidentiality, all data is made available by contacting the PI, James Stavitz, at jstavitz@gmail.com or jstavitz@kean.edu. Data will be made available upon request.

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
