# Peer review of "Exploring Individual Mental Health Issues: A Qualitative Study among Fellowship-Trained Sports Medicine Physicians"

_ijerph, 2023, doi:10.3390/ijerph20075303_

Round 1

Reviewer 1 Report

Thank you for the opportunity to review your manuscript. In general, there are some formatting issues to align with IJERPH guidelines including the author lines and information for the associated institution. In addition, level headers should adhere to journal guidelines with numbering and structuring as required by IJERPH template. Finally, the citation formatting should have in-text citation in parentheses followed by the period.

Abstract – The abstract should be unstructured and 200 words. Please restructure. Change to past tense (is to was on line 10). Consider changing males and females to men and women.

Key points – these can be removed. Not required for the journal.

Line 29: Need to redefine FTSMP in the manuscript. This should be done in addition to the abstract.

Line 30: Covid-19 should be all caps (COVID-19) and consider spelling out completing the first time

Line 32: An effort on the toll? I am not sure this makes sense as the authors go on to describes issues rather than efforts.

Introduction – In general, the introduction may benefit from some explanation of what a FTSMP is, their role, and why we should care about the phenomenon being studied. There is no background on why this population was studied.

Line 49 – citation needed

Line 53 – citation needed

I would suggest making Lines 54-65 a Table versus setting up in this format.

The introduction should end with how a gap in the literature has been established and your purpose statement or aims. This must be revised.

Methods

1.      Move Line 74-75 to end of the introduction

2.      Avoid starting sentences with abbreviations (Line 81, 91, etc.)

3.      Line 83 – need to add comma before C

4.      How did the researchers establish B prior to the interview?

5.      Line 91 – The abbreviation has already been introduced and not needed again

6.      In would be interesting for the study to identify where/why the 40 screening surveys were excluded. What were they missing?

7.      Change male/female to man/woman to align with self-reported gender rather than biological sex.

8.      Ensure all methods are past tense (Line 104 – is to was)

9.      Table 1 – move California over to align with over states

10.   Line 111 – not sure “followed” is the best word here

11.   Line 112 – How was the signature obtained when this was done via Qualtrics?

12.   Line 113 – what do you mean by “lead the study”

13.   A line needs to be added to the methods about IRB approval/review

14.   Figure 2 needs to be reformatted – it is very confusing, the text does not fit in all of the circles well and seems distracting more than helpful.

15.   Table 2 – question 2 and 3 has grammatical issues that should be addressed

16.   I would suggest adding some citations around constant-comparative coding

17.   Line 139 – unswerving is an interesting word choice – what do you mean by it?

18.   Line 144 – why is there a citation here?

19.   Can you help to explain more about who did the coding? Was it the PI? It did not indicate the research team role outside of reviewing the protocol.

20.   Line 149 – citation needed

21.   Table 4 is unnecessary and should be removed

22.   The ethical consideration section is very helpful, but not necessary for a published manuscript in the detail provided. Consider streamlining this section.

Results

23.   Table 5 – rather than number of excerpts, it may be best to show this as number of participants (out of 35) who made comments about the theme.

24.   If you created pseudonyms, why were they not used in the paper? I noticed in Table 6 you used P9, P2, etc. This may be improved by using their pseudonym.

25.   Table 6 – All quotes should be cleaned for grammatical purposes

26.   Theme 2, P1 quote – how does this actually match the theme?

27.   There is a lack of context to how the quotes are provided. For example, there are time where it seems they are responding to a direct question (Theme 2, P5 and P6) and without you providing context to the quote, it is lost in the table format.

28.   Table 6 – It concerns me that some participants are represented multiple times and others are never represented. If you have 35 unique participants, I think it is key that more of them are represented in the paper.

29.   Theme 4 – the quotes do not make sense for the theme. How does P6 and P5 explain poor help seeking behaviors. This is a big concern.

30.   The results are quite long with the quotes in the table and the text.

Dicussion

31.   The discussion would benefit from a general overview of the findings and then leading into the STS.

32.   Change Line 356 to include stress

33.   Line 397 – I would remove this as it is results information rather than discussion

34.   I think the discussion would be improved to have a section on each of the themes or sub-headers to align with each of the main theme results

35.   Line 448 – this is not a small sample size. Remove from the limitations

36.   Line 470 – galvanized does not seem to be the best word choice here

Missing the author contributions, funding, IRB statement, informed consent statement, data availability statement, acknowledgments, an conflicts of interest.

Reviewer 2 Report

Exploring Individual Mental Health Issues: A Qualitative 2 Study among Fellowship-Trained Sports Medicine Physicians

The manuscript deals with an important topic and should definitely be considered as useful for clarifying some health issues. It addresses the mental health of sports medicine physicians, which is important as this profession is important for providing healthcare. Also, it is original and could make a good basis for further examining the personal mental health of other occupations. The manuscript lacks some explanations about the topic and should be technically amended. Therefore, I have some comments and suggestions to improve the manuscript.

Abstract should be unstructured, which means that you should remove Objective, Design…

Affiliations are not correctly stated. Use the number and provide the full address and e mail addresses of the authors.

INTRODUCTION

You should introduce what FTSMP is at the beginning of the introduction. Abbreviations used in the abstract should also be introduced in the main text as those are two distinguished parts of the manuscript. Also, a more detailed purpose and role of FTSMPs should be described.

Covid-19 should be COVID-19; please correct.

You should describe why and how the COVID-19 pandemic has affected people in general and FTSMPs' job and life.

I believe that you used the wrong in cite reference type throughout the manuscript. Same goes for the Reference list at the end of the manuscript.

Line 45: I believe there is no need (and perhaps incorrect for this journal) to have a new title of the paragraph (theoretical foundations). I think you should merge it in introduction. Also, please do not leave the numbering of factors like that; just separate them by a comma in a sentence-like manner (exposure to suffering, empathic concern, empathic ability…). This way, the manuscript should be more fluent.

You are lacking research questions, aims and hypothesis at the end of the introduction. But, before adding it you should add one paragraph that combines FTSMP and STS and COVID-19 and why it should be important. Of course, supported by references of previous studies.

METHODS

I believe methods are not in line with the journal's requirements.

Did you explain to FTSMP what mental issues are and which ones they might be having? Was there a chance that they did not understand what mental health and mental issue are? (this is based on the Table 2 Interview questions)

Line 195: Please state what IRB stands for.

FIGURES AND TABLES

Tables and figures are a bit messy; please arrange them properly. (e.g., figure 3., table 4)

DISCUSSION

Line 357-358: Please add reference(s)

Line 376: Please remove this Title (Interpretation) as Discussion should be interpretation of the results so this is redundant.

Line 402: Reference is missing.

The conclusion is well written and gives a systematic overview of the whole study. Also, it states how the results could be used to improve the situation regarding FTSMP mental health.

Round 2

Reviewer 1 Report

Thank you for the time completing the revisions. I do feel the edits have helped with the context and flow of the paper. Some of the responses seemed to ask for changes upon request.

Would the reviewer feel more
comfortable with the section if the research team added the questions that were being
asked for the participant to record such a response - Yes

But over-quoting (i.e. including too many quotes, or quotes that are too long) is a very common problem - Agreed. As a qualitative researcher, I can appreciate your sentiments. However, it does appear some participants are referenced in the results more than others. With 28-33 participants being represented, I am suggesting some variability for those selected - not more quotes.

Table - I like the table. It is just long and loses context to how the quote fits at times. My suggestion is to streamline it not remove.

Still formatting issues - please look at the MDPI IJERPH template.

Reviewer 2 Report

The authors have done a substantial work for improving thier manuscript. I consider this manuscript very well written now. 

 I believe there is no need (and perhaps incorrect for this journal) to have a new title of the paragraph (theoretical foundations). I think you should merge it in introduction. Also, please do not leave the numbering of factors like that; just separate them by a comma in a sentence-like manner (exposure to suffering, empathic concern, empathic ability…). This way, the manuscript should be more fluent. Response: Reviewer 1 had a similar comment. However, they noted that the 12-Factor Model should be in table form. Would the current reviewer feel more comfortable with the items listed and separated by a comma.

You can leave it in a table form, it looks ok now in the manuscript.
